# The Genome Analysis of the Human Lung-Associated *Streptomyces* sp. TR1341 Revealed the Presence of Beneficial Genes for Opportunistic Colonization of Human Tissues

**DOI:** 10.3390/microorganisms9081547

**Published:** 2021-07-21

**Authors:** Ana Catalina Lara, Erika Corretto, Lucie Kotrbová, František Lorenc, Kateřina Petříčková, Roman Grabic, Alica Chroňáková

**Affiliations:** 1Institute of Soil Biology, Biology Centre Academy of Sciences of The Czech Republic, Na Sádkách 702/7, 37005 České Budějovice, Czech Republic; ana.lara@bc.cas.cz (A.C.L.); erika.corretto@gmail.com (E.C.); lucie.kotrbova@upb.cas.cz (L.K.); lorenc.frantisek@gmail.com (F.L.); 2Institute of Immunology and Microbiology, 1st Faculty of Medicine, Charles University, Studničkova 7, 12800 Prague 2, Czech Republic; katerina.petrickova@lf1.cuni.cz; 3Faculty of Science, University of South Bohemia, Branišovská 1645/31a, 37005 České Budějovice, Czech Republic; 4Faculty of Fisheries and Protection of Waters, University of South Bohemia, Zátiší 728/II, 38925 Vodňany, Czech Republic; rgrabic@frov.jcu.cz

**Keywords:** *Streptomyces*, human lungs, respiratory tract, lung pathogenic actinomycetes, comparative genomics, adaptations to human tissue colonization, secondary metabolites, biosynthetic gene clusters

## Abstract

*Streptomyces* sp. TR1341 was isolated from the sputum of a man with a history of lung and kidney tuberculosis, recurrent respiratory infections, and COPD. It produces secondary metabolites associated with cytotoxicity and immune response modulation. In this study, we complement our previous results by identifying the genetic features associated with the production of these secondary metabolites and other characteristics that could benefit the strain during its colonization of human tissues (virulence factors, modification of the host immune response, or the production of siderophores). We performed a comparative phylogenetic analysis to identify the genetic features that are shared by environmental isolates and human respiratory pathogens. The results showed a high genomic similarity of *Streptomyces* sp. TR1341 to the plant-associated *Streptomyces* sp. endophyte_N2, inferring a soil origin of the strain. Putative virulence genes, such as mammalian cell entry (*mce*) genes were not detected in the TR1341’s genome. The presence of a type VII secretion system, distinct from the ones found in *Mycobacterium* species, suggests a different colonization strategy than the one used by other actinomycete lung pathogens. We identified a higher diversity of genes related to iron acquisition and demonstrated that the strain produces ferrioxamine B in vitro. These results indicate that TR1341 may have an advantage in colonizing environments that are low in iron, such as human tissue.

## 1. Introduction

Actinomycetes have become one of the most relevant groups of microorganisms since the discovery of antibiotics in the 1940s. They are gram-positive bacteria with high GC content, and comprise taxa known to be producers of a wide range of secondary metabolites. The *Streptomyces* genus is considered as the most relevant genus, since it has become the source of about two-thirds of all known natural antibiotics [1,2].

Even though streptomycetes are predominantly known as soil- and sediment-associated microorganisms, they have also been reported as free-living sea water bacteria [3,4], and have been found in extreme habitats [5,6,7,8] and as symbionts of plants [9], invertebrates [10,11,12,13], and turtles [14]. In most cases, their interaction with the host has been classified as mutualistic because streptomycetes protect the host against fungal or bacterial infections [15]. Nevertheless, some *Streptomyces* species cause diseases in plants through the production of phytotoxins and other specific metabolites. *Streptomyces scabies*, for instance, is the main causative agent of potato scab, which affects the roots and the tubers of the potato plant [16,17,18].

Due to their ability to produce antibacterials, antifungals, and immunomodulators as secondary metabolites, *Streptomyces* are generally viewed as a bioresource by humans. There are only a few *Streptomyces* species that are pathogenic under certain conditions: *Streptomyces somalensis* and *Streptomyces sudanensis*, which cause endemic actinomycetoma, for example [19,20].

Generally speaking, the main reservoir for human-associated *Streptomyces* strains is the soil. At the same time, contact with soil is an important factor in human health [21]. It is therefore not so farfetched to consider *Streptomyces* strains as potential control agents in the developing microbial communities as possible trainers of the human immune system [22,23]. *Streptomyces* species have been reported to inhabit healthy skin [24], the uterus [25], the gastrointestinal system [22], and the respiratory tract [26,27]. They are also suspected of mediating communication between the gut microbiota, the brain, and the lungs [28].

Albeit mostly soil dwellers, streptomycetes (and especially their spores) may be air-borne. Particularly in highly polluted areas with high particulate matter content, such as barns with hay storage, wet moldy houses, urban and industrial smog, or dusty air after soil and sand storm events [29,30,31]. Consequently, their spores can be inhaled easily and may start to germinate. It was shown that even at this development stage, *Streptomyces coelicolor* starts the production of secondary metabolites that not only control their growth but are also involved in signaling and communication with other organisms [32]. In vitro, *Streptomyces* was one of the main producers of cytotoxic and inflammatory compounds present in mold-infested houses [33]. *Streptomyces californicus*, along with *Stachybotrys chartarum*, are commonly isolated from water-damaged or mold-infested buildings together. It is hypothesized that these two organisms could be responsible for respiratory tract problems in patients living in water-damaged buildings [34]. So far, there are no reports of *Streptomyces californicus* isolation from lung tissue or sputum. However, its spores, in combination with *Stachybotrys chartarum*, induced an inflammatory response in macrophages of mice [35] and produced cytostatic compounds with immunotoxic properties [36]. Other clinical reports linked *Streptomyces* to pulmonary infections, abscesses of different organs, and bacteremias found mostly in immunocompromised patients and in a few immunocompetent individuals [37]. The limited data about the colonization of human tissues by non-pathogenic *Streptomyces* has recently been summarized by Herbrík et al. [27].

The human-colonization strategies of pathogenic actinomycetes are best studied in mycobacteria, which use mechanisms different to other bacterial pathogens. Host-adapted mycobacteria, such as *Mycobacterium tuberculosis*, do not employ virulence factors and toxins. Unlike other pathogenic bacteria, they have evolved specific strategies to outsmart the immune system and exploit the resources of the host [38]. A recently discovered mechanism used by actinomycetes for host immune evasion, modulation, and exploitation are the specialized Type VII secretion systems (T7SS), also called ESX [39]. Studied mostly in *Mycobacterium* but found also in *Streptomyces*, *Corynebacterium*, *Nocardia*, and *Gordonia*, ESX secretion systems allow the interaction of the infectious bacteria with the host immune system [40]. *Mycobacterium tuberculosis* has five paralogous T7S systems (ESX-1–ESX-5). ESX-1, together with horizontally acquired *espACD* loci, mediates the rupture of phagosomal membranes inside the host phagocytes. ESX-3 acts in metal ion homeostasis. ESX-5 induces caspase-independent cell death and facilitates the spread of the infection by modulating the innate immune response and inhibiting the production of cytokines in type I macrophages [41,42]. ESX-4 induces immunogenicity in mice characterized by significant lymphocyte proliferation and induction of TNF-α and IL-6. It is also the smallest and apparently the most ancestral ESX system in the genus *Mycobacterium*, with the other ESX systems evolving from it by gene duplication events and insertion. The putative function of ESX-2 remains unknown. So far, studies on *Streptomyces scabies* and *Streptomyces coelicolor* proved that their ESX system seems to be involved in sporulation and development rather than virulence [43,44].

Another important colonization mechanism is mediated by mammalian cell entry (MCE) proteins. They are a family of invasion proteins described in mycobacteria and present in some other *Actinomycetales*. MCE proteins allow nonpathogenic *E. coli* cells to enter and multiply within non-phagocytic cells [45]. The presence of *mce* genes in a genome does not guarantee the pathogenicity of the strain, but it highly correlates with it [45,46,47]. However, the deletion of the *mce* cluster in *Streptomyces coelicolor* A3(2) allowed for rapid germination of the spores inside the macrophage model *Acanthamoeba polyphaga*, but reduced the efficiency of colonization of the roots of *Arabidopsis thaliana* [46]. The enhanced virulence associated with a non-functioning *mce* operon was also observed by Shimono et al., who reported an increase in virulence in a mutant with a disruption in the *mce-1* operon of *M. tuberculosis* [48]. Thus, these results suggest that if present before, the loss of the *mce* operon might increase the virulence of a given strain.

Once in the airways or lungs, microorganisms need a supply of essential trace elements, mostly iron, which in turn may unbalance the iron homeostasis of the host [49]. Invading microbes scavenge iron by secreting small molecules called siderophores. Siderophores have a large chemical structural diversity and are widely produced by streptomycetes [50]. They can chelate available iron ions, but also steal iron from iron-binding proteins produced by the human body, or by other members of the microbiome. This can lead to an antibiotic effect [51,52,53]. Siderophores can also be covalently linked to antibiotics forming sideromycins [54]. They are popular antibiotics because of the dual mechanism in which they function and most of them are produced by actinomycetes (albomycins, ferrimycins, danomycins, etc.) [51]. Other antibiotics produced by actinomycetes that also have metal biding characteristics are: streptomycin and novobiocin that bind Cu^2+^, erythromycin that binds Co^2+^, and tetracycline that binds Ca^2+^ and Fe^2+^ [55].

*Streptomyces* sp. TR1341 was isolated from the sputum of a senior man, who was living in the mining region of Příbram, Czechia, and had a history of lung and kidney tuberculosis (TB), recurrent respiratory infections, and chronic obstructive pulmonary disease (COPD). In our previous study we showed that TR1341’s antifungal, β-hemolytic, and cytolytic activities were due to the production of filipin III and fungichromin, and the antibacterial and cytotoxic activities were due to the production of actinomycin X_2_. In vitro experiments showed that *Streptomyces* sp. TR1341’s germinating spores reduce viability and activation of human macrophages. This could protect the strain upon its entrance into the human body. Co-culture of the strain with a human macrophage line further induces the cytotoxic activity [27].

In this study, we would like to address the following questions: (i) how does TR1341’s whole genome relate to other pathogenic actinomycetes? (ii) what mechanism of host evasion/colonization can we predict from TR1341 genome? (iii) does TR1341 possess genes or biosynthetic clusters that are commonly present in known pathogenic actinomycetes, but are absent in “environmental” strains?

In order to answer these questions, we performed a detailed analysis of the strain’s genome and compared it with 140 genomes of pathogenic and non-pathogenic actinomycetes from different environments. Here we show that *Streptomyces* sp. TR1341 has a potentially better adaptation strategy to the human tissue niche which involves the production of specific secondary metabolites, or other abovementioned colonization mechanisms. These mechanisms give TR1341 an edge over other environmental strain.

## 2. Materials and Methods

### 2.1. Streptomyces sp. TR13141: Isolation Method, Cultivation, and Genome Sequencing

*Streptomyces* sp. TR1341 was isolated from the sputum of an 81-year-old male patient with a complex clinical history including repeated infections of the respiratory tract. The sputum sample was subjected to selective cultivation for mycobacteria as described in Hebrík et al. [27]. The strain was kept in The National Reference Laboratory for Pathogenic Actinomycetes in the Local Hospital in Trutnov, Czech Republic, which kindly provided it the purposes of our research.

We routinely cultivated *Streptomyces* sp. TR1341 at 28 °C on M2 (malt extract 10 g, yeast extract 4 g, glucose 4 g, agar 20 g, distilled water 1 L). In order to investigate the effect of temperature on its growth, we also grew TR1314 at 37 °C, on M2 and tryptic soy agar (TSA) plates.

Details of the genome sequencing and assembly can be found in Hebrík et al. [27]. Briefly the library was prepared using the TruSeq PCR free LT library preparation kit (Illumina) and sequences with the Illumina MiSeq platform (Reagent kit v2, paired end, 300 bp). After polishing the reads with Trimmomatic 0.36 [56], the assembly was performed using FLASH [57] and SPAdes [58], and the quality of it was evaluated with QUAST [59] and Qualimap [60]. The draft genome (VSDL00000000.1) was annotated using Prokka v. 1.14.5 [61].

### 2.2. Morphological Characteristics and Microscopy

All the culture characteristics of the strain were determined through standard methods given in the International Streptomyces Project (ISP) [62]. The strain was streaked onto four different media: ISP2 (malt extract 10 g, yeast extract 4 g, dextrose 4 g, agar 20 g, distilled water 1 L), ISP3 (HiMedia; oat meal 20 g, FeSO_4_ × 7H_2_O 0.001 g, MnCl_2_ × 4H_2_O 0.001 g, ZnSO_4_ × 7H_2_O 0.001 g, agar 18 g, distilled water 1 L), M2 (malt extract 10 g, yeast extract 4 g, glucose 4 g, agar 20 g in 1 L of distilled water, pH 7.2) and MS (mannitol 20 g, soy flour 20 g, agar 20 g, tap water 1 L) [63]. The plates were incubated at 28 °C for 2 weeks, and colony morphology, signature characteristics of the spore coloration, and sporophore formation were observed.

The morphology of the sporophores produced by TR1341 that was grown on MS media for 14 days was observed with the scanning electron microscope JEOL 7401-FE (JEOL Ltd., Tokyo, Japan) at the accelerating voltage of 4 kV. The sample (aerial mycelium) was fixed and dehydrated in vapors of OsO_4_, then frozen and, after evaporation, coated with gold using a Sputter Coater (Baltec-SCD 050).

The spore surface was classified by the examination of carbon replicas of spores with the transmission electron microscope (TEM Jeol 1010) at the acceleration voltage of 80 kV with Mega View III camera (SIS).

### 2.3. Growth Curve and Carbon Utilization Assay

Strain TR1341 was pre-cultivated on M2 agar for 7 days at 28 °C and 37 °C. Because we wanted to use McFarland turbidity for standardizing of the inoculum, a suspension was prepared by vigorous vortexing of the mycelia and spore mixture in a small volume of sterile Luria-Bertani (LB) broth (tryptone 10 g, yeast extract 5 g, NaCl 10 g in 1 L of distilled water, pH 7.0) in a sterile microcentrifuge tube with glass beads. The supernatant was used to prepare a suspension of 0.5 McFarland turbidity (densitometer DEN-1, Biosan, Latvia). A total of 100 µL of suspension was inoculated into 25 mL of LB broth and cultivated at 28 °C and 37 °C in an orbital shaker (150 rpm, Multitron Standard, Infors AG, Bottmingen-Basel, Switzerland), in baffled 100 mL Erlenmeyer flasks supplemented with approximately 30 sterile glass beads (average 0.3 mm) to achieve a dispersed growth. OD_600_ measurements were carried out using a NanoDrop One spectrophotometer (Thermo Fisher Scientific, Waltham, MA, USA) at regular intervals. The experiment was performed in triplicates. The same conditions were maintained for the determination of the growth rate of *Streptomyces nodosus* ssp. *asukaensis* ATCC 29757, which was chosen as a reference strain due to its homogeneous growth (no cell aggregates) in liquid media.

Carbon utilization was evaluated using sterile carbon sources and following the protocol given in the International Streptomyces Project (ISP) [62].

### 2.4. Siderophore Production and Identification

For the detection of siderophore production, a modified chromeazurol S (CAS) assay, using the dual agar plate method, was performed with malt extract agar as the cultivation medium and CAS agar as the detection medium [64,65]. *Streptomyces* sp. TR1341 was grown at 28 °C on MS agar plates to promote sporulation [63]. One loop of active spores was inoculated on the malt extract agar side at a 0.5 cm distance from the CAS agar side. The production of siderophores was indicated by the change of color from blue to red/orange in the CAS agar part. The zone was monitored and measured for two weeks, after which the area showing a color change, and thus containing the secreted siderophores (width ~ 2.0 cm), was cut. The siderophores were extracted by adding 2.5 mL of acetonitrile:water:formic acid solution (1:1:0.1 *v*/*v*/*v*) and shaking at 140 rpm 4 °C for 2.5 h. Experiments and extractions were performed in triplicates.

Siderophores were also extracted from freeze-dried biomass from 100 mL cultures (GYM medium, 150 rpm, 72 h, 28 °C) by adding 20 mL of acetonitrile:water:formic acid solution. These liquid assays were done by duplicates. Crude extracts were filtered with Chromafil^®^ Regenerated cellulose syringe filters before analysis (pore size 0.45 μm, diam. 30 mm).

The filtered extracts were analysed using reversed-phase liquid chromatography with electrospray ionization and high-resolution mass spectrometry. A volume of 10 µL of the extract was injected first in the same phase precolumn (10 mm × 2.1 mm ID 5 µm particles), and then onto the analytical column (HypersilGold aQ, 50 mm length, 2.1 mm ID with 5 µm particles, Thermo Scientific, Waltham, MA, USA). A gradient of acetonitrile in water (acidified with 0.1% formic acid) was used for the separation of the analytes. The starting composition of the mobile phase was 100% of water at a flow of 0.35 µL·min^−1^ for 1 min, then the acetonitrile content was linearly increased to 25% in 4 min, 60% in 8 min, 100% in 10 min, followed with 2 min isocratic 100% acetonitrile.

The ion source conditions were as follows: vaporizer temperature 300 °C, capillary temperature 325 °C, sheath gas 40 arb. units, auxiliary gas 12 arb. Units, and spray ionization voltage 2800 and 3500 V for negative and positive ionizations modes, respectively. The QExactive HF (Thermo Fisher Scientific, Waltham, MA, USA) hybrid quadrupole/orbital trap high resolution mass spectrometer was used for detection in a combined full scan/data independent MS^2^ experiment.

One run was performed in positive ionization mode, the other in negative ionization mode. Full scan data were acquired in the range of 100–1300 *m*/*z* with an ion time of 100 ms. DIA was performed with an isolation window of 100 *m*/*z* for *m*/*z* 150, 250, 350, 450, 550, and 650 with an ion time of 70 ms, stepped collision energy 15, 30, and 50 NCE, and a 30,000 resolution of product scan. Data were processed and evaluated using Xcalibur and MassFrontier 7.1 software (ThermoScientific).

Full scan chromatograms were searched against an *m*/*z* list of known desferrioxiamines and ferrioxiamines published in literature [66,67]. A maximal difference from the theoretical mass of 5 ppm was set for this search.

### 2.5. Dataset

In addition to the genome of *Streptomyces* sp. TR1341, a total of 140 genomes were downloaded from the GenBank database (as of September 2020). The genomes were carefully selected based on genome quality and representativeness of a wide variety of isolation sources such as soil, rhizosphere, manure, water, air, marine invertebrates, and insects (Appendix A). First, we selected the *Streptomyces* species clustering with *Streptomyces* sp. TR1341 on the 16S rRNA gene tree calculated in Hebrík et al. [27]. Next, we added well-studied organisms like *Streptomyces albus* DSM 41398, *Streptomyces avermitilis* MA-4680, and *Streptomyces coelicolor* A3(2). We also included *Streptomyces albidoflavus* NBRC 100770, *Streptomyces daghestanicus* NRRL B-2710, *Streptomyces diastaticus* subsp. *ardesiacus* NBRC 15402, *Streptomyces koyangensis* VK-A60T, *Streptomyces flavogriseus* ATCC 33331, *Streptomyces seoulensis* KCTC 9819, and *Streptomyces xiamenensis* MCCC 1A01550. Based on 16S rRNA gene analysis, these strains showed similarity to *Streptomyces* isolated from sputum samples collected in the same area of TR1341 (data not published yet). In order to investigate the potential role of TR1341 as a pathogenic agent, we included all the available genomes of human-associated *Streptomyces* species. We also selected 20 genomes of known human (lung-)pathogens having less than 200 contigs and affiliated to the following taxa: *Actinomadura madurae*, *Gordonia*, *Mycobacterium*, *Nocardia asteroides*, *Rhodococcus,* and *Tsukamurella* (Appendix A).

We filtered our initial database by discarding all genomes with more than 300 contigs and then we evaluated genome completeness using BUSCO v. 3.0.2 [68]. Genomes that failed to satisfy the requirement of >90% completeness and overall quality of 50% were discarded. Exceptions were made when the low-quality assembly belonged to a pathogenic or reference species with no better assembly available (e.g., *Streptomyces somaliensis* DSM 40738).

All genomes were re-annotated using Prokka v. 1.14.5 [61] to avoid variability due to the annotation method in downstream analysis.

### 2.6. Orthologous Genes and Species Tree Inference

Groups of orthologous sequences were identified and phylogenetic inference was done using the abovementioned 141 genomes with the software OrthoFinder v. 2.4.0 [69]. OrthoFinder uses not only single copy orthologous genes, but also the full set of orthologous genes for phylogenetic inference. Due to the algorithm used for tree inference (STAG), the branch lengths correspond to the average number of substitutions per site across a large range of gene families [70]. The resulting tree is therefore a species tree that can be used directly in downstream analysis such as ancestral state reconstruction and calibration.

### 2.7. Biosynthetic Gene Clusters

Gene clusters were identified using antiSMASH v 5.1.2 set to the “relaxed” detection strictness option in order to detect complete clusters, as well as partial clusters, in which some genes might be missing in fragmented draft genomes [71]. For this analysis, we used a subset of genomes as reported in Appendix A. All cluster predictions for TR1341 were further analyzed in PRISM [72].

The Antibiotic Resistant Target Seeker (ARTS 2.0) [73] was used to search for potential novel antibiotic targets in the genome of *Streptomyces* sp. TR1341.

### 2.8. Mammalian Cell Entry (mce) Genes and Type VII Secretion System (ESX)

Using the Batch CD-Search [74], we identified the main conserved protein domains of the *mce* genes: MlaE (PF02405.16) and MlaD (PF02470.20). In order to detect the type VII secretion system genes, we identified the conserved protein domain of five essential genes: WXG100 for *esxA* and *esxB* (PF06013.12), T7SS_ESX1_EccB for *eccB* (PF05108.13), FtsK_SpoIIIE for *eccC* (PF01580.18), YukD for *eccD* (PF08817.10), and EccE for *eccE* (PF11203.8). We downloaded the corresponding profiles from the Pfam database [75] and used them to search for *mce* and type VII secretion system genes in a subset of the selected genomes, using Hmmer v. 3.2 [76].

### 2.9. Genomic Comparison of Streptomyces sp. TR1341, Streptomyces sp. Endophyte_N2, and the Human Associated Streptomyces spp.

We ran two sets of comparisons: one set was done using the information produced by Orthofinder [69]; a second using the functional groups identified by RAST [77]. Shortly, we ran an analysis of amino acid sequence similarity between TR1341 and *Streptomyces* sp. Endophyte_N2, *Streptomyces brasiliensis* strain NRRL B-1626 (human associated, no clear pathogen), *Streptomyces somaliensis* DSM 40738 (human pathogen), and *Streptomyces* sp. KE1 (human associated, healthy skin).

Using the RAST annotation server, we identified genes associated with the following functional groups: antibiotic resistance, virulence factors, iron acquisition, acyl carrier proteins, phospholipases, glyoxylate, and mobile elements (phages and prophages).

To visualize the results of these comparisons, we constructed Venn diagrams using Venny 2.1.0 [78].

## 3. Results and Discussion

### 3.1. Streptomyces sp. TR1341: Morphological Characterization and Genome Features

*Streptomyces* sp. TR1341 was isolated from the sputum of an elderly man, who was treated for several respiratory tract-related diseases for decades. The strain produces aerial mycelium with spores that are smooth and form spiral sporophores (Appendix A).

We evaluated TR1341 growth rate at 28 °C and at 37 °C, and observed that TR1341 reaches the maximum OD faster at 37 °C (day 22) than at 28 °C (day 30). The reference strain, on the other hand, had a higher OD measurement at 28 °C (Appendix A).

When growing at 28 °C, TR1341 produced an orange pigment in the substrate mycelia on both M2 and TSA (color of substrate mycelium, ISP coding: Oc4r-Coo4b-4s/Grayish brownish orange), but the aerial spores developed completely only on M2. Cultivated at 37 °C on TSA, however, TR1341 does not produce an orange pigment, nor does it develop aerial mycelia. When grown at 37 °C on M2, the strain produces a small amount of orange pigment in the substrate mycelium and the typical pigmentation of aerial spores (ISP coding: GY5fe—Light grayish reddish brown) is partly inhibited (Appendix A).

The carbon utilization test (Appendix A) was negative for L-arabinose, sucrose, rhamnose, raffinose, and cellulose. The results for D-xylose, I-inositol, and D-fructose were inconclusive. Interestingly, TR1341 can use both mannitol and glucose which is the positive control. Mannitol is commonly used in the treatment of patients with obstructive pulmonary disease. Inhaled mannitol (commercial name Bronchitol [79]) reduces the inflammation in the airways and improves mucociliary clearance [80,81]. Unfortunately, we lack information on the patient clinical history and were therefore unable to confirm a link between the use of mannitol as part of the patient’s treatment and its possible subsequent use as a carbon source by TR1341.

The draft genome of *Streptomyces* sp. TR1341 has a total length of 8.5 Mb and is divided into 170 contigs. It has a GC content of 71.8% and 7442 coding sequences, as reported in Table 1. The genome size and its GC content are comparable to the *Streptomyces* genomes selected for this study, which have an average genome size of 8.3 ± 1.5 Mb (Appendix A). Among the selected *Streptomyces*, 16 have a genome shorter than 7 Mb. Two out of the three human-associated *Streptomyces* are part of this group: *Streptomyces somaliensis* DSM 40738 (5.2 Mb) and *Streptomyces* sp. KE1 (6.8 Mb). The other strains were isolated from marine habitats (6), plants (3), rhizospheres (3), and soil (2). Besides *Streptomyces somaliensis* DSM 40738, *Streptomyces xiamensis* is the only strain with a genome shorter than 6 Mb, despite the fact that it was isolated from mangrove sediments [82]. Another study shows that *Streptomyces* derived from marine sponges are remodeling their genomes by gene acquisition or loss in order to better adapt to different hosts [83]. When considering the plant pathogens of *Streptomyces* species, it appears that only a few of them have a reduced genome. For instance, the pathogens *Streptomyces albidoflavus* NBRC 13083 has a genome of 7 Mb (Appendix A), whereas *Streptomyces scabiei* S58, *Streptomyces turgidiscabies* T45, and *Streptomyces acidiscabies* have much bigger genomes (around 10 Mb) [84].

Taken all together, it seems that *Streptomyces* strains keep a relatively large set of genes as a genome foundation and then they might add, silence or lose functions while colonizing a new environment and adapting to it. This flexibility allows them to survive in a wide range of environments. For instance, *Streptomyces somaliensis* DSM 40738 can live and survive either in the human body or the soil, despite having a reduced genome compared to the average *Streptomyces* genome size. Since these *Streptomyces* did not undergo a drastic genome reduction that forces them to steal the deleted function from the host (e.g., mycoplasma), we can conclude that these organisms, including TR1341, have a lifestyle that is different to that of obligate pathogens.

### 3.2. Orthologous Genes and Tree Inference

We identified 26,770 orthologous groups in the set of 141 strains. These were used to construct the tree and show the relationships between the strains. As expected, *Streptomyces* form a separate cluster from the other actinomycetes, with *Actinomadura madurae* being the closest taxon (Figure 1).

*Streptomyces* sp. TR1341 is located among the soil- and plant-derived strains, a large distance from other human-associated strains (*Streptomyces somaliensis* DSM 40738, *Streptomyces brasiliensis* NRRL B-1626, and *Streptomyces* sp. KE1) and as expected, far away from other pathogenic actinomycetes. The closest organism is *Streptomyces* sp. endophyte_N2 (Figure 1), with whom TR1341 shares more than 95% identity at amino acid level throughout the genome (Appendix A). This is also true for *Streptomyces costaricanus* DSM 41827, the closest clustering strain in the analysis made by Hebrík et al. [27]. *Streptomyces* sp. endophyte_N2 was isolated from *Arabidopsis thaliana* roots and has a broad antimicrobial activity [85]. It shares 5113 orthologous genes with TR1341 and has only 170 unique orthologous genes compared to the 293 that are present in TR1341 (Figure 2a). It seems that each of the four human-associated *Streptomyces* falls into a different cluster of the tree (Figure 1). This could reflect the fact that pathogenic and human-associated *Streptomyces* are principally opportunistic pathogens and therefore the closest related organisms in the tree are the ones from the primary habitat. For the moment, this remains a hypothesis that future studies could address with the addition of new publicly available genomic data of other *Streptomyces* isolated from human samples.

For a deeper comparative genomics analysis, we arbitrarily chose 19 strains from our dataset: eight strains that closely cluster with TR1341 on the species tree (soil-derived and endophytes), the three human-associated strains of *Streptomyces*, three *Mycobacterium* strains, one *Gordonia,* and two *Tsukamurella* strains, all associated with lung tissue as human pathogens.

### 3.3. Biosynthetic Gene Clusters

The software antiSMASH was used to predict secondary metabolite gene clusters. *Streptomyces* sp. TR1341 harbors 42 clusters. Among these, 13 had more than 50% of the genes showing similarities to already known biosynthetic gene clusters (BGCs) (Table 2). A total of 26 clusters were confirmed using PRISM, whereas 16 and 8 clusters were predicted only by antiSMASH and PRISM, respectively (Appendix A).

The most represented category is non-ribosomal protein synthetases (NRPS), followed by polyketide synthases (PKS). In our previous study, we already proved that TR1341 is able to produce filipin and actinomycin X_2_ [27]. The filipin BGC is similar to the one characterized in *Streptomyces avermitilis* NRRL 8165 [86] and *Streptomyces filipinensis* DSM 40112 [87], from which it was first isolated as an antifungal agent [88]. It belongs to the group of polyene antibiotics and it is highly toxic for human cells. In TR1341, it is responsible for its β-hemolytic activity, providing the strain an advantage over fungal competitors and probably enhancing the release of nutrients from the host cells. Besides the loss of β-hemolytic activity, the deletion of the key PKS genes in the filipin cluster causes an increase in the production of actinomycin X_2_, a derivative of actinomycin D [27]. The actinomycin BGC of TR1341 shows similarity to the one of *Streptomyces chrysomallus* ATCC 11523 [89]. We were able to identify both compounds, actinomycin D and actinomycin X_2_, in the extracts of the submersed cultures (GYM media, samples 1 and 2). Like *Streptomyces antibioticus* IMRU 3720 [90], TR1341 has only one biosynthetic arm and the predicted substrates of the actinomycin synthetases (ACMS) are threonine-valine (ACMS II) and proline-glycine-valine (ACMS III). Moreover, in TR1341, the *acmN* gene, encoding a ferredoxin, is replaced by mobile elements, suggesting that this function might be carried out by a ferredoxin located outside the cluster. In the context of adaptation to the human body, the highly antibiotic and cytotoxic properties of actinomycin provide TR1341 with another tool to control gram-negative competitors and scavenge nutrients.

Other useful metabolites for the interaction of TR1341 with eukaryotic cells are hopene and rhizomide (Table 2). Hopene is the precursor for the synthesis of pentacyclic triterpenoids called hopanoids [91]. Besides their role in membrane stabilization [92], different studies demonstrated that they can act as anti-leukemia, anti-inflammatory, and anti-oxidation natural products [93,94]. Rhizomides have weak antitumor activity and protect against plant diseases like cucumber downy mildew. Their biosynthetic pathway has been characterized in *Paraburkholderia rhizoxinica* HKI 454 [95]. Even though three clusters of TR1341 show similarity to BGC for rhizomide A/B/C (Table 2), they might represent the same region. In fact, they are located on short contigs, which might have failed to assemble into a longer one due to the repetitions of the long NRPS gene involved in the synthesis of rhizomides (MIBiG accession BGC0001758).

The concentration of essential elements such as iron is strictly regulated inside the human body and the role of molecules such as siderophores is therefore essential. *Streptomyces* sp. TR1341 seems to be able to synthesize the siderophores mirubactin [96] and desferrioxamine [97] through a NRPS-dependent and -independent pathway, respectively (Table 2). We qualitatively confirmed the production of siderophores in vitro via the CAS agar test (Appendix A). In the crude extracts, we could not detect the siderophore mirubactin, but we were able to identify ferrioxamine B using reversed-phase liquid chromatography with electrospray ionization and high-resolution mass spectrometry (delta ppm from theoretical *m*/*z* −0.984 and isotope pattern corresponds to 1 Fe within the molecule). The structure was confirmed by finding a match with published MS/MS spectra fragments. In addition, we identified a homologue that has a very similar retention time but with a shorter chain (one CH_2_ less). Both compounds showed an identical low mass fragment while heavier fragments were specific for the given compound (Figure 3a). XIC at M^+^ peak areas of the extracts are reported in Figure 3b to document the semi-quantification of the produced compounds. The high reproducibility of the experiment and analysis is documented by the almost identical peak areas for the triplicates of the cultivation on dual CAS agar plates (MEA/CAS, TR1341/1–3), and the very similar peak areas for the duplicates of the submersed cultures (TR1431/1–2).

In addition, one of TR1341’s clusters shows similarity to the BGC of diisonitrile antibiotic SF2768 identified in several *Streptomyces* species and other *Actinobacteria*. This molecule mediates the uptake of copper and acts as an antifungal and virulence factor [98]. In fact, besides their role in nutrient uptake, siderophores and other chelators act as antibiotic agents sequestering essential elements and preventing other microorganisms from access to these resources [99,100].

When looking at the BGC, in which at least 50% of the genes show similarity to a known cluster, we observed that most of TR1341’s BGCs are also present in the soil/plant-associated *Streptomyces* spp. belonging to its phylogenetic cluster (Figure 4). *Streptomyces costaricanus* DSM 41827, *Streptomyces* sp. endophyte_N2, and *Streptomyces griseofuscus* NG1-7 have almost the same predicted BGCs. For instance, previous studies reported the production of actinomycin and filipin in *Streptomyces costaricanus* SCSIO ZS0073 and *Streptomyces* sp. endophyte_N2, respectively [85,101]. The geosmin BGC is present in all selected *Streptomyces* spp. apart from *Streptomyces somaliensis* DSM 40738. On the other hand, the cluster for the osmoprotectant ectoine is widespread among *Streptomyces* [102] and it is the only BGC shared by TR1341, *Mycobacterium smegmatis* MC2-155, and the lung-associated actinomycetes such as *Gordonia bronchialis* DSM 43247, *Tsukamurella pulmonis* DSM 44142, and *Tsukamurella tyrosinosolvens* NCTC13231 (Figure 4). Only five of TR1341’s BGCs of TR1341 are present in the human-associated *Streptomyces*: desferrioxamine, diisonitrile antibiotic SF2768, geosmin, hopene, and melanin.

In most cases, the number of predicted BGCs per organism correlates with the size of the genome: the larger the genome, the higher the number of predicted clusters (Figure 5, Appendix A). It seems that the human-related *Streptomyces* strains are in an intermediate position between the lung-pathogen actinomycetes and the soil/plant-related *Streptomyces*. Indeed, smaller genomes are a sign of adaptation to a symbiotic/pathogenic lifestyle compared to community. For instance, the genome size of *Streptomyces somaliensis* DSM 40738 is comparable with that of the lung-pathogen actinomycetes, but it has more BGCs. These might still provide metabolites that are beneficial for survival in the soil before the colonization of human tissues via skin lesions. Therefore, the transition of *Streptomyces somaliensis* DSM 40738 to the human environment from the soil might have happened before the transition of other human-associated *Streptomyces*, such as TR1341. In any case, for *S. somaliensis* and TR1341, it seems that the human environment is just another possibility in terms of habitat and we do not have any indication of niche specificity.

To complement the antiSMASH analysis, the Antibiotic Resistant Target Seeker (ARTS) was used to search for potential novel antibiotic agents in the genome of *Streptomyces* sp. TR1341 [73]. A total of 587 genes (about 7% of the total) were identified as core/essential genes and 41 known resistance models were found. The major four functional categories of the core genes are: protein synthesis (18%), biosynthesis of cofactors, prosthetic groups, and carriers (12%), energy metabolism (12%), and unclassified (10%) (Appendix A).

Among these, 14 genes satisfied at least three of the four criteria set by ARTS: duplication, proximity to BGC, phylogeny/horizontal gene transfer, and similarity to known resistance models. Regarding the proximity to a BGC, 24 clusters satisfy this criterium, having at least one core gene hit (Appendix A). For instance, the BGC Region 20.1 corresponding to cluster 19 in ARTS contains eight core genes, but only 22% of the genes show similarity to the known BGC kinamycin. The genes showing similarity are not the main biosynthetic ones. In fact, the predicted PKS and NRPS genes seem to show similarity to the genes for the synthesis of other compounds like cystothiazole A and myxothiazol. In BGC Region 34.1 (cluster 27 in ARTS), we identified four genes and a resistance model related to a biotin-requiring enzyme, even though only 6% of the genes showed similarity to the formicamycins A-M BGC (Table 2, Appendix A).

These examples show that even BGCs exhibiting a low percentage of similarity to previously characterized clusters are worth investigating. ARTS provides the information that enables the prioritization of research into unknown BGCs, which may lead to the discovery of new bioactive molecules.

### 3.4. Mammalian Cell Entry (mce) Genes and Type VII Secretion System (ESX)

We could not detect the presence of genes encoding the conserved domains MlaD and MlaE, which characterize the *mce* genes (Table 3) in the genome of *Streptomyces* sp. TR1341.

Among the selected *Streptomyces* strains, only *Streptomyces* sp. KE1 and *Streptomyces brasiliensis* NRRL B-1626 harbor one complete *mce* operon. They were both isolated from human samples but are not clearly associated with pathogenicity.

On the other hand, all the other actinomycetes harbor multiple *mce* operons, with *Mycobacterium simiae* MsiGto having the highest number (nine operons). Interestingly, the *Streptomyces somaliensis* DSM 40738 genome does not possess any *mce* operon, indicating that this strain uses a different virulence strategy than the typical strategy employed by actinobacterial lung pathogens. The fact that *Streptomyces somaliensis* DSM 40738 is not a lung pathogen but affects skin, soft tissues, and bones, probably relates to the differences that we observe in the infection strategies at this level (*mce* genes). A *Streptomyces somaliensis* DSM 40738 infection is characterized primarily by an inflammatory reaction [103], which is more related to the presence of type VII secretion systems (ESX) and not to *mce*-encoded proteins. Unlike *Mycobacterium* species, where all five ESX secretion systems are widespread and sometimes even duplicated [42,104], all the selected *Streptomyces* genomes, with the exception of *Streptomyces* sp. KE1, harbor at least one ESX system similar to the one characterized in *Streptomyces scabies* 87.22 and *Streptomyces coelicolor* A3(2) (Figure 6, Appendix A).

The fact that this ESX system has a different structure than the one present in pathogenic mycobacteria and that it is present in several *Streptomyces* isolated from different sources suggests that it indeed might be involved in developmental processes instead of virulence.

The complete ESX system in TR1341 is more similar to that of *Streptomyces coelicolor* A3(2) than to that of *Streptomyces scabies* 87.22 (Figure 6). In the plant pathogen *Streptomyces scabies* 87.22, *esx*A and *esx*B do not have the characteristic WXG100 protein family domain, but instead have an FNG-motif and FQA-motif, respectively. Moreover, a subtilisin-type serine protease distant from the ESX cluster might act as the transmembrane protein EccB [43,44].

*Streptomyces* sp. endophyte_N2, the *Streptomyces griseofuscus* strains (NG1-7, NRRL B-5429, 64), *Streptomyces malaysiense* MUSC 136, and *Streptomyces brasiliensis* NRRL B-1626 had a second copy of the system, where *eccE* was missing (Appendix A). In *Streptomyces* sp. TR1341, *Streptomyces murinus* NRRL B-2286, and *Streptomyces* sp. LamerLS-31b, this second copy is lacking the two WXG100-Esx genes probably due to the fact that they are located near the edge of a contig. Other genes from the WXG100 family were found in other parts of the genomes. *Gordonia bronchialis* seems to have all the genes necessary for a functional ESX system, but they are scattered along the whole genome. *Tsukamurella pulmonis* DSM 44142 harbors two ESX systems, whereas in *Tsukamurella tyrosinosolvens* NCTC13231 we could detect all the domains located on different plasmids. The presence of ESX-encoding plasmids has been reported before in mycobacteria and is considered to play a major role in the spreading and diversification of type VII systems [105].

The lack of *mce* genes and a type VII secretion system that is distinct from the one found in pathogenic mycobacterium species suggest an alternative invasion and colonization strategy in TR1341, the nature of which is unfortunately still not clear.

### 3.5. Genomic Comparison of Streptomyces sp. TR1341, Streptomyces sp. Endophyte_N2, and the Human Associated Streptomyces spp.

To the best of our knowledge, only three genomes of human-associated *Streptomyces* are deposited in the GenBank database (September 2020). *Streptomyces somaliensis* DSM 40738 is the causative agent of actinomycetoma and shows the genetic adaptation of a pathogenic lifestyle (i.e., small genome size) [20]. *Streptomyces brasiliensis* NRRL B-1626 and *Streptomyces* sp. KE1 were isolated from “human disease” [106] and healthy human skin surface (SAMN03454257, unpublished), respectively. *Streptomyces* sp. TR1341 shares the highest number of orthologous gene families with *Streptomyces brasiliensis* NRRL B-1626 (3671), followed by *Streptomyces* sp. KE1 (3400) and *Streptomyces somaliensis* DSM 40738 (2945) (Figure 2b). These numbers reflect the distances showed in the phylogenetic tree in Figure 1.

In order to easily assign the identified orthologous genes to specific functional categories, we used RAST [77] to annotate the genomes of *Streptomyces* sp. endophyte_N2 and the three human-associated *Streptomyces*. Results of the comparison at protein level showed that TR1341 and *Streptomyces* sp. endophyte_N2 share 99% of sequence identity along most of the genome, with the exception of two small regions that have a similarity of around 60% (henceforth referred to as the red region) and 80% (the blue region) (Appendix A).

The high similarity at the amino acid level indicates that perhaps these two strains have originated from a similar environmental niche and so respond to similar evolutionary constrains [107]. The red region (Appendix A) is a low similarity region in the central part of the chromosome and is composed of a few annotated genes and a group of hypothetical proteins. We cannot predict any specific biological activity from the group of annotated genes and antiSMASH failed to predict any BGCs in this region. However, these genes might be involved in a biosynthetic pathway, since most of them are synthetases or transferases, and there is a LysR transcriptional regulator.

The blue region is at the end of the left arm of the genome (Appendix A) and it is unique to TR1341, not appearing in any of the other human-associated *Streptomyces*. Most of the genes present in TR1341 code for hypothetical proteins and those with an assigned function are annotated as mobile elements, phage/prophage-associated genes, or are involved in the mobilization of genetic material (recombinational DNA repair, DNA helicase). These kinds of genes are often found at the distal arms of the *Streptomyces* chromosome representations [108].

After, we performed a thorough analysis on the following genes and functional categories: antibiotic resistance and virulence, iron metabolisms, phospholipases, and glyoxylate pathway and mobile elements.

Regarding the antibiotic resistance genes and virulence factors, TR1341 has four genes, which are not found in the human-associated *Streptomyces* (Figure 7a, Appendix A). The first of these genes is annotated as a putative internalin (RAST) from marine actinobacterium PHSC20C and *Flavobacterium johnsoniae,* or as hypothetical protein (Prokka, NCBI). Internalin is a surface protein, which is well-studied in *Listeria monocytogenes* and is involved in the invasion of mammalian cells via cadherins transmembrane proteins and Met receptors (hepatocyte growth factor receptor, HGFR) [109]. The putative internalin in TR1341 has very low amino acid similarity with the sequence of the internalin gene from *Listeria*, which could indicate a very different function in TR1341, or it could be that the differences in sequences are not translated to differences in 3D structure. However, at the moment, we have no way of testing any of these hypotheses. The amino acid sequence in TR1341 contains a DUF11 domain that has been associated with porin formation in *Chlamydia trachomatis,* and so it is regarded as important for the pathogenicity of the bacteria. The sequence DUF11 is widely distributed and well conserved among *Streptomyces* genomes of environmental origin (Appendix A). The second gene unique to TR1341 is the streptothricin acetyltransferase (*Streptomyces lavendulae* type); this gene provides resistance to streptothricin [110]. The other two unique genes belong to the family of MerR transcriptional regulators.

Our previous study showed that TR1341 is resistant to penicillin, ampicillin, cephalosporin, quinolones, rifampicin, tetracycline, and trimethoprim/sulfamethoxazole [27]. Here, we classified the genes associated with these resistance profiles (Appendix A). TR1341 has 13 genes associated with resistance to penicillin, ampicillin, and cephalosporin. Most of these genes are part of the class C beta-lactamases. Four genes are involved in the strain’s resistance to fluoroquinolones: two topoisomerase and two DNA gyrases. We found two genes for the strain’s resistance to rifampicin, three genes for the strain’s resistance to tetracycline, and three genes for the strain’s resistance to trimethoprim-sulfamethoxazole. These final three are ABC transporters.

In the RAST category for iron acquisition, there are four genes unique to TR1341 when compared to the other human-associated *Streptomyces* (Figure 7b, Appendix A). These genes are unique forms of segments of the EfeUOB system; specifically, two unique forms of *EfeU* and *EfeO*. EfeU serve as integral membrane iron permeases, whereas the role of EfuO is not entirely known but it could act as an iron-binding or an electron-transfer component [111].

These results, coupled with the positive result of siderophore production assay and the presence of siderophore BGCs (Appendix A, Table 2), suggest that TR1341 has a wider variety of iron acquisition mechanisms compared to other strains in our analysis. This high efficiency could confer TR1341 with an advantage for colonizing low-iron environments such as the lungs.

When inside the human body, microbes are in contact with macrophages as a primary line of defense against infection. Studies showed that when captured by macrophages, the yeast *Saccharomyces cerevisiae*, the opportunistic fungal pathogen *Candida albicans,* and the lung pathogen *Mycobacterium tuberculosis* activate the glyoxylate shunt in order to use even-chain fatty acids as a carbon source [112,113,114]. Interestingly, in heterotrophic bacteria, the glyoxylate shunt plays an important role under iron deficient conditions [115]. The two enzymes of the glyoxylate shunt, isocitrate lyase and malate synthase, are present in *Streptomyces* sp. TR1341 and *Streptomyces brasiliensis* NRRL B-1626, but could not be found in the other selected genomes (Appendix A).

Other enzymes that influence the interaction of microbes with eukaryotic cells are phospholipases and sphingomyelinases [116]. TR1341 harbors only phospholipases C and lysophospholipases, whereas the other *Streptomyces* strains also have phospholipases D and sphingomyelinases (Appendix A). These enzymes are produced by both gram-positive and gram-negative bacteria and some can act as virulence factors. For instance, they hydrolyse membrane lipids that release nutrients essential for colonization, and in some cases they disrupt the immune response by affecting the homeostasis of the host’s signaling molecules [117,118].

It comes as no surprise that mobile elements are far more variable among the analyzed strains (Figure 7c, Appendix A). There are nine sequences in this subsystem that are unique to TR1341. Most of them (seven) have ambiguous annotations (mobile element protein, phage related protein, phage protein). Only two have better understood functions and are annotated as the recombinational DNA repair protein RecT (prophage-associated) and as a transposase.

## 4. Conclusions

Our results show that TR1341 has an increased growth rate at 37 °C when compared to 28 °C. It produces the siderophore ferrioxamine B and can use mannitol as a carbon source. These physiological characteristics can be considered as an adaptation to the human niche.

The phylogenetic analysis shows that *Streptomyces* sp. TR1341 is quite distant from both human-associated and pathogenic strains, even though it was isolated from a sputum sample. We did not observe a genome reduction, which is usually associated with obligate pathogens. We could predict a large number of BGCs and we were able to identify several genes associated with the scavenging of nutrients (e.g., iron), which are not present in other human-associated *Streptomyces*. The various genes related to antibiotic resistance may be the result of the frequent antibiotic treatments that the patient underwent and could thus have been acquired after colonization. On the other hand, the diversity of ABC transporters present in TR1341 could point to an innate advantage of the strain related to antibiotic resistance. The lack of *mce* genes and the presence of a type VII secretion system distinct from the one employed by *Mycobacterium tuberculosis* species suggest a different colonization strategy.

All these results imply a greater metabolic plasticity of TR1341 when compared to other typical human-associated streptomycetes. A highly plastic metabolic capacity confers the microorganism with the ability to grow in many different environments, including the human lungs. This high potential for metabolic plasticity is also a characteristic of soil-associated bacteria when compared to specialized pathogens because soil-associated bacteria need to survive in highly unstable environments. On the other hand, specialized pathogens tend to discard the information that is not needed in their new and “stable” environment. Moreover, we detected more gene variants in TR1341 (e.g., multiple slightly different copies of the same gene) than in the other pathogenic strains. This could be an adaptive response to a constantly changing environment, and could also point to the fact that TR1341 is not a specialized pathogen but an opportunistic one. To sum up, it seems that TR1341 was a soil-dwelling strain that accidentally entered the lungs of the patient and had enough genetic mechanisms at its disposal to allow it to not only survive, but to thrive inside the patient’s lungs.

This thorough genomic analysis can be used as a road map for future experiments that will improve understanding of the specific colonization strategies of the respiratory tract by TR1341 and further advance our knowledge of the role of streptomycetes in the human body.

## Figures and Tables

**Figure 1 microorganisms-09-01547-f001:**
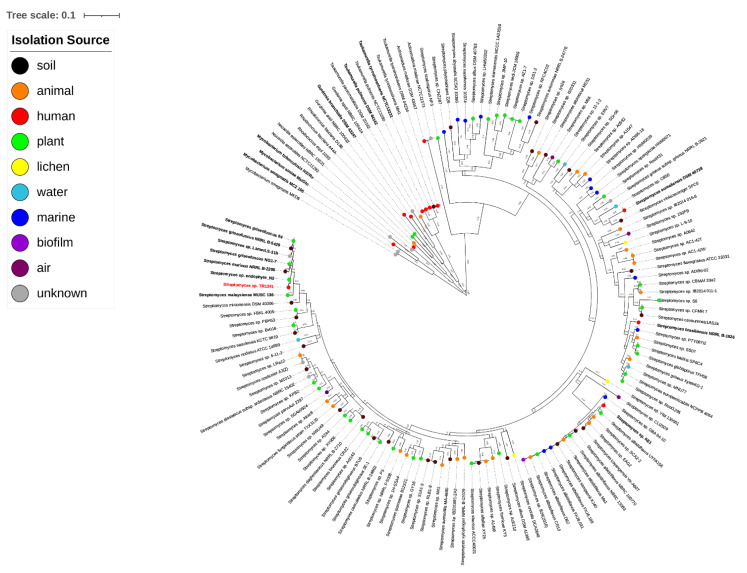
Phylogenetic tree based on orthologous families calculated with Orthofinder. Branch lengths are indicated.

**Figure 2 microorganisms-09-01547-f002:**
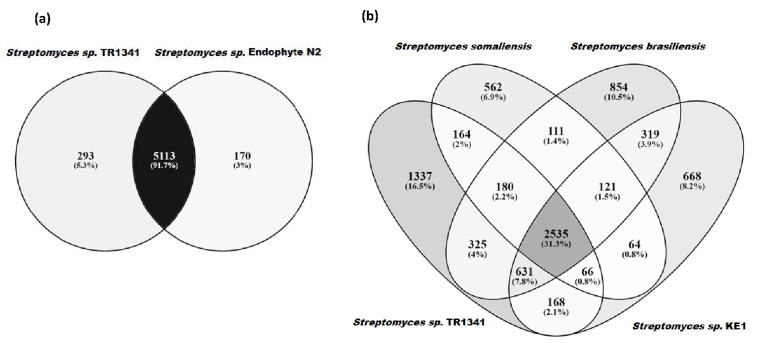
Comparison of orthologous groups in (**a**) *Streptomyces* sp. TR1341 with the closely related *Streptomyces* sp. endophyte_N2, and (**b**) the human-associated *Streptomyces* species: *Streptomyces somaliensis* DSM 40738, *Streptomyces brasiliensis* NRRL B-1626, and *Streptomyces* sp. KE1.

**Figure 3 microorganisms-09-01547-f003:**
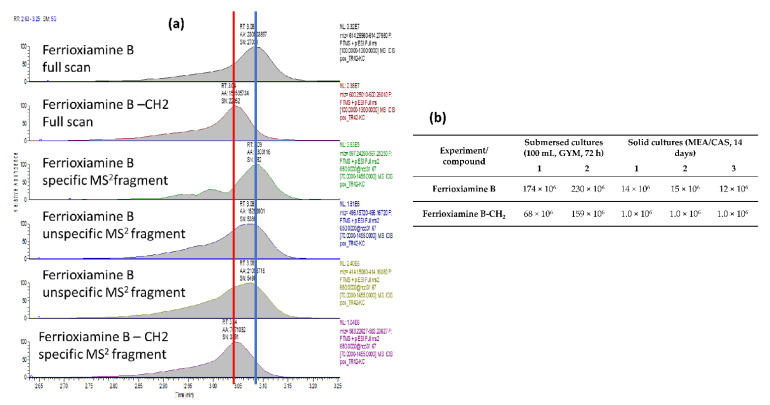
Ferrioxiamine B and its -CH_2_ homologue detected in the extracts from the submersed cultures (1–2) and from the solid cultures (1–3). (**a**) Extracted XIC chromatograms of ferrioxiamine B and its -CH_2_ homologue. (**b**) Semiquantitative evaluation of ferrioxiamine B and its homologue.

**Figure 4 microorganisms-09-01547-f004:**
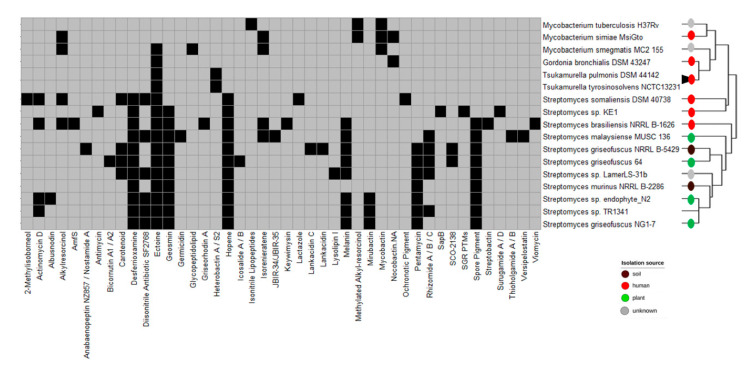
Biosynthetic gene clusters (BGCs) predicted by antiSMASH with more than 50% of the genes showing similarity to a known BGC. Black squares indicate presence, while grey indicates absence. The isolation source of the organisms is shown with the colored circles at the tip of the branches.

**Figure 5 microorganisms-09-01547-f005:**
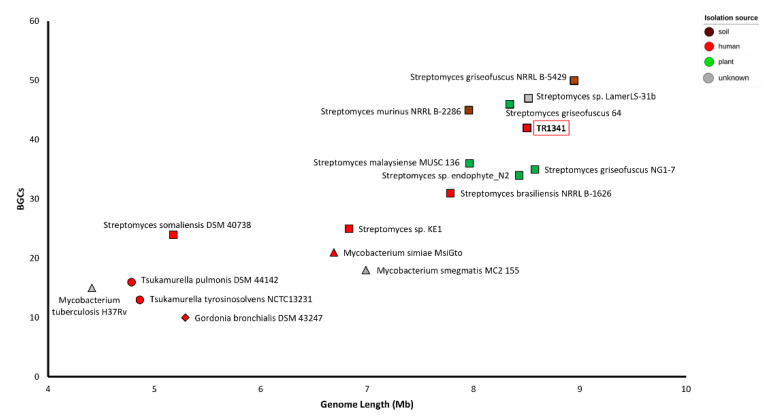
Relationship between genome size (Mb) and number of biosynthetic gene clusters (BGCs). Squares: *Streptomyces* species; triangles: *Mycobacterium* species; circles: *Tsukamurella* species; diamond: *Gordonia bronchialis* DSM 43247.

**Figure 6 microorganisms-09-01547-f006:**
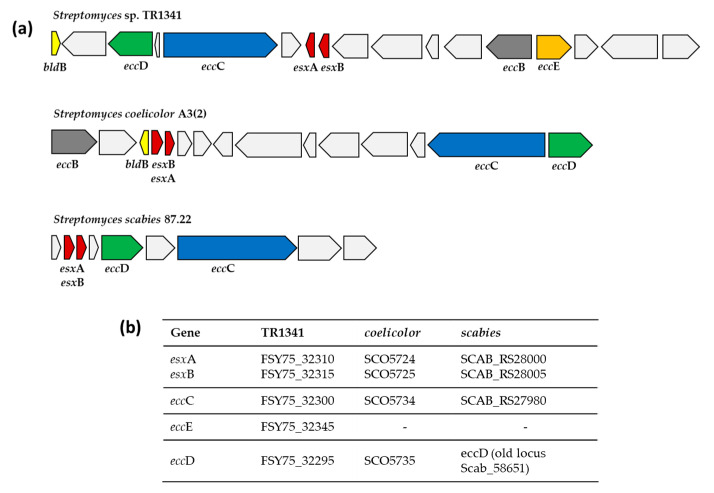
Comparison of type VII secretion system (ESX) present in *Streptomyces* sp. TR1341 and the ones characterized in *Streptomyces coelicolor* A3(2) and *Streptomyces scabies* 87.22. (**a**) Schematic representation of the genomic region of the different type VII secretion systems. (**b**) Genomic locus tags of the main genes.

**Figure 7 microorganisms-09-01547-f007:**
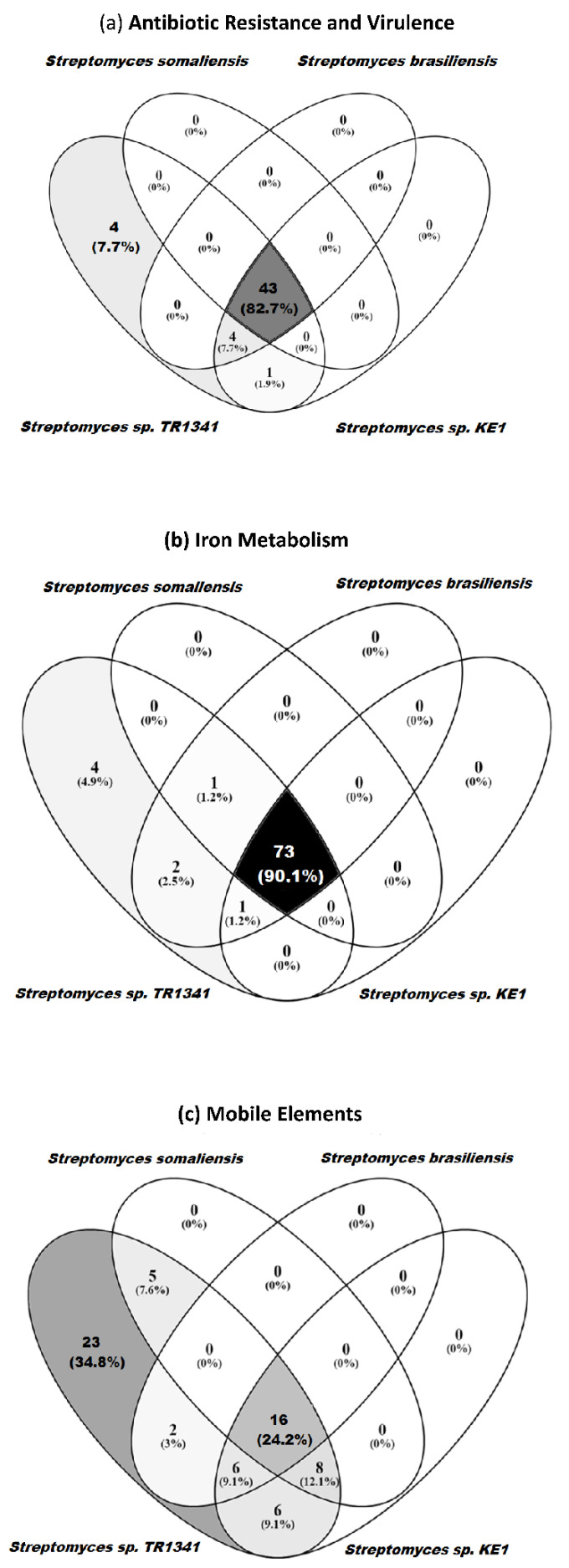
Genes shared between *Streptomyces* sp. TR1341 and the human-associated *Streptomyces* species in the following RAST categories: (**a**) proteins associated with antibiotic resistance and virulence factors; (**b**) proteins associated with iron metabolism; (**c**) mobile elements (associated to secondary metabolism but no other subsystem).

**Table 1 microorganisms-09-01547-t001:** Genomic features of *Streptomyces* sp. TR1341, and comparison with *Streptomyces* sp. endophyte_N2 and *Streptomyces somaliensis* DSM 40738.

	*Streptomyces* sp. TR1341	*Streptomyces* sp. Endophyte_N2	*S. somaliensis* DSM 40738
Genome size (Mb)	8,507,620	8,428,700	5,176,903
Contigs	170	1	243
GC content (%)	71.80	71.83	74.20
CDS (Prokka)	7442	7253	4613
rRNA (5S/16S/23S)	7/8/3	7/7/7	6/6/6
tRNA	90	90	70
tmRNA	1	1	1
CRISPR	3	3	1

**Table 2 microorganisms-09-01547-t002:** Biosynthetic gene clusters (BGCs) predicted in *Streptomyces* sp. TR1341 using antiSMASH [66].

BGC Type	BGC	Most Similar Known BGC	Genes Showing Similarity (%)
bacteriocin	Region 29.1	Informatipeptin	28
	Region 6.2	-	-
betalactone, PKS-like	Region 5.2	bafilomycin B1	11
ectoine	Region 67.1	Ectoine	100
hglE-KS	Region 39.1	Primycin	5
lanthipeptide	Region 23.1	-	-
melanin	Region 84.1	Melanin	60
NRPS	Region 116.1	rhizomide A/rhizomide B/rhizomide C	100
	Region 139.1	rhizomide A/rhizomide B/rhizomide C	100
	Region 15.1	Mirubactin	78
	Region 1.2	diisonitrile antibiotic SF2768	66
	Region 76.1	Stenothricin	18
	Region 55.1	acyldepsipeptide 1	15
	Region 2.1	Bleomycin	9
	Region 25.1	caniferolide A/caniferolide B/caniferolide C/caniferolide D	4
	Region 30.2	-	-
	Region 109.1	-	-
NRPS, betalactone	Region 17.1	Kirromycin	16
	Region 34.1	formicamycins A–M	6
NRPS, ectoine	Region 3.1	Showdomycin	23
NRPS, other	Region 11.1	actinomycin D	89
NRPS, siderophore	Region 30.1	salinosporamide A	16
NRPS, T1PKS	Region 1.3	Pentamycin, Filipin	100
NRPS, T1PKS, transAT-PKS-like	Region 2.2	cinnabaramide A	18
NRPS-like	Region 162.1	rhizomide A/rhizomide B/rhizomide C	100
	Region 18.1	Paulomycin	13
NRPS-like, T1PKS	Region 1.1	Borrelidin	9
other, lanthipeptide	Region 10.2	A-503083 A/A-503083 B/A-503083 E/A-503083 F	7
PKS-like, T1PKS, other	Region 6.3	Meilingmycin	5
siderophore	Region 64.1	Desferrioxamine	66
T1PKS	Region 77.1	Catenulisporolides	3
T1PKS, NRPS	Region 112.1	-	-
T1PKS, NRPS, terpene	Region 10.1	Ebelactone	5
T1PKS, siderophore, NRPS	Region 20.1	Kinamycin	22
T2PKS, T1PKS	Region 13.1	spore pigment	83
T3PKS	Region 27.1	Herboxidiene	7
T3PKS, NRPS	Region 92.1	A-47934	26
terpene	Region 6.1	Geosmin	100
	Region 26.1	julichrome Q3-3/julichrome Q3-5	25
	Region 24.1	-	-
	Region 43.1	-	-
terpene, thiopeptide, LAP	Region 5.1	Hopene	92

**Table 3 microorganisms-09-01547-t003:** Summary of Mammalian Cell Entry (MCE) genes presence in the studied genomes. MleD (Pfam PF02470.20): conserved domain of the two ABC-transporter integral membrane proteins, yrbEA; MleE (Pfam PF02405.16): conserved domain of the six hypothetical proteins, mceABCDEF.

Origin	Organism	Strain	No. MleD	No. MleE	No. Mce Operons
Soil/plant	*Streptomyces* sp.	TR1341	-	-	-
	*Streptomyces* sp.	endophyte_N2	-	-	-
	*Streptomyces griseofuscus*	NG1-7	-	-	-
	*Streptomyces griseofuscus*	NRRL B-5429	-	-	-
	*Streptomyces griseofuscus*	g64	-	-	-
	*Streptomyces malaysiense*	MUSC 136	-	-	-
	*Streptomyces murinus*	NRRL B-2286	-	-	-
	*Streptomyces* sp.	LamerLS-31b	-	-	-
Human	*Streptomyces* sp.	KE1	6	2	1
	*Streptomyces brasiliensis*	NRRL B-1626	6	2	1
	*Streptomyces somaliensis*	DSM 40738	-	-	-
Human	*Mycobacterium tuberculosis*	H37Rv	24	8	4
	*Mycobacterium smegmatis*	MC2-155	38	12	6
	*Mycobacterium simiae*	MsiGto	56	16	9
Human	*Gordonia bronchialis*	DSM 43247	29	9	5
Human	*Tsukamurella pulmonis*	DSM 44142	25	8	4
	*Tsukamurella tyrosinosolvens*	NCTC13231	19	6	3

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
