# Peer review of "The Genome Analysis of the Human Lung-Associated Streptomyces sp. TR1341 Revealed the Presence of Beneficial Genes for Opportunistic Colonization of Human Tissues"

_microorganisms, 2021, doi:10.3390/microorganisms9081547_

Round 1

Reviewer 1 Report

The manuscript ‘The genome analysis of the human lung associated Streptomyces sp. TR1341 revealed the presence of beneficial genes for opportunistic colonization of human tissues” is follow up study of the same strain in which detailed analysis of the strain genome is presented with focus on the colonization of the human host. In the manuscript authors has employed number of bioinformatic tools and comparative genome analysis of the strain with other Actinobacteria strain, both pathogenic and environmental. They have performed phylogenetical analysis, examined strain ability to produce siderophores, investigated strain biosynthetic potential by biosynthetic gene cluster identification, check the potential for antibiotic resistance, looked for the genes connected with virulence. The authors look more specifically on the organization of the genes for two system involved in pathogenicity of Actinobacteria, specifically mammalian cell entry (mce) and type VII secretion system (ESX). The study is thoughtfully designed, includes number of adequate analysis and well curated dataset of strains for comparative analysis considering the scope of the paper. It is important addition to the previous paper by the same group focusing on TR1341 strain.

On the strong sides of the paper the number of genome analysis of use of more that one tool for some have to be pointed out. That is important as the algorithms are not perfect and cannot be treated as an oracle, so employing more than one tool for particular analysis can provide stronger support for the conclusions. Comparative analysis has been made on the well curated set of strains that were selected in relevance to the theme of the paper, e.g. inclusion of other pathogenic Streptomyces for comparative analysis even though they were not closely related phylogenetically to the TR1341 strain.

There is few, rather minor points and suggestions for changes that might improve paper readability:

  • As for the comparative analysis sets of strains was well curated, why the authors used strains clustered based only on 16S rRNA gene for the phylogenetic analysis? It is quite established currently that 16S rRNA is not enough for phylogenetic analysis or even pre-selection of the strain, especially in the genera where vast number of strains are sequenced.
  • In Material and Methods section, a general sentence or two about sequencing of the strain genome could be added. The authors refer to their previous paper which is good, however just mentioning method and sequencing approach could be included, and for more details people could go to previous paper.
  • Lines 306-307; the sentence about spores, seems like random inclusion, with no further explanation why it is important or if it has any effect in the context of the paper. I would suggest either removing it or providing more detail on overall strain phenotype (e.g. colony shape, growth description, pigmentation).
  • Lines 360 – 363 – statement that the TR1341 in phylogenetical analysis is located far away from other human associated Streptomyces is true. But at the same time all of other human associated Streptomyces are separated from each other, so it is not very specific for the TR1341, that should be clearly stated in the manuscript. For the other pathogenic Actinobacteria, based on dataset it is also not uncommon that the strain is separated from them, as they belong to different genera and just few strains of these genera were included which is the cause they were grouped. Perhaps changing the data presentation on Figure 1 so it shows evolutionary distance would improve the quality of the figure and message it delivers.
  • Line 366 – ‘…made by [27].’ I suggest changing it to ‘…made by Herbrik, et al. [27].’
  • Line 397 – it is not clear to me what the information in brackets is refereeing too, specifically ‘…(GYM media, samples 1 and 2)’ – please specify that in the text.
  • Line 398 – what the author meant by ‘…has only one biosynthetic arm…’?
  • Line 415 – what the number in brackets refers to, is it the contig mentioned in the sentence or reference sequence of NRPS involved in the synthesis of rhizomides in another strains, specify please.
  • Lines 594 – 596 – authors state that low sequence similarity points to different function which in my opinion is a slight overstatement. It might be true, however does not have to be as the function of proteins is determined by theirs structure, and even though the sequence defines structure, there is still a possibility that proteins with low sequence similarity will have highly similar structure. That is usually the case of the proteins under strong evolutionary pressure, e.g. in vertebrates immune system. If the protein is involved in the pathogenicity of the strain, it would be reasonable to highly diversify its structure, so the chances to be recognized by the host immune system are minimized. I suggest rephrasing the sentence so it discusses on the options and is less suggestive.

Author Response

The manuscript ‘The genome analysis of the human lung associated Streptomyces sp. TR1341 revealed the presence of beneficial genes for opportunistic colonization of human tissues” is follow up study of the same strain in which detailed analysis of the strain genome is presented with focus on the colonization of the human host. In the manuscript authors has employed number of bioinformatic tools and comparative genome analysis of the strain with other Actinobacteria strain, both pathogenic and environmental. They have performed phylogenetical analysis, examined strain ability to produce siderophores, investigated strain biosynthetic potential by biosynthetic gene cluster identification, check the potential for antibiotic resistance, looked for the genes connected with virulence. The authors look more specifically on the organization of the genes for two system involved in pathogenicity of Actinobacteria, specifically mammalian cell entry (mce) and type VII secretion system (ESX). The study is thoughtfully designed, includes number of adequate analysis and well curated dataset of strains for comparative analysis considering the scope of the paper. It is important addition to the previous paper by the same group focusing on TR1341 strain.

On the strong sides of the paper the number of genome analysis of use of more that one tool for some have to be pointed out. That is important as the algorithms are not perfect and cannot be treated as an oracle, so employing more than one tool for particular analysis can provide stronger support for the conclusions. Comparative analysis has been made on the well curated set of strains that were selected in relevance to the theme of the paper, e.g. inclusion of other pathogenic Streptomyces for comparative analysis even though they were not closely related phylogenetically to the TR1341 strain.

There is few, rather minor points and suggestions for changes that might improve paper readability:

  • As for the comparative analysis sets of strains was well curated, why the authors used strains clustered based only on 16S rRNA gene for the phylogenetic analysis? It is quite established currently that 16S rRNA is not enough for phylogenetic analysis or even pre-selection of the strain, especially in the genera where vast number of strains are sequenced.

Answer: The phylogenetic tree shown in Figure 1 has been calculated with Orthofinder, which considers all orthologous gene families shared by the selected genomes. In our previous paper (Herbrík et al., 2020), we performed a phylogenetic analysis based on the 16S rRNA gene (Figure 1 in the reference). This tree was supported by a second phylogenetic tree calculated using 83 single-copy genes in autoMLST. 16S rRNA gene is broadly available for various strains for which no genome of expected quality is not available so far, this is the reason, why we used 16S rRNA as a first marker to search for similar strains. As a next step, we used different criteria for selection as described in M&M (l. 255-267).

  • In Material and Methods section, a general sentence or two about sequencing of the strain genome could be added. The authors refer to their previous paper which is good, however just mentioning method and sequencing approach could be included, and for more details people could go to previous paper.

Answer: A brief description of the methods used for genome sequencing and assembly has been added to the materials and methods section 2.1. Streptomyces sp. TR13141: isolation method, cultivation and genome sequencing (l.166-170).

  • Lines 306-307; the sentence about spores, seems like random inclusion, with no further explanation why it is important or if it has any effect in the context of the paper. I would suggest either removing it or providing more detail on overall strain phenotype (e.g. colony shape, growth description, pigmentation).

Answer: In this paragraph, we provide a general characterization of TR1341 morphology and physiology. For this purpose, we used different media (M2, TSA) and incubated the strain at different temperatures (28-37°C). We noted that the different conditions influenced both the pigment production and the formation of spores. For this reason, we included a sentence about the spore morphology in this section. However, the spore formation is general feature, typical for the strain, not being influenced by culture media and the sentence was thus rewritten (l.443-444).

  • Lines 360 – 363 – statement that the TR1341 in phylogenetical analysis is located far away from other human associated Streptomyces is true. But at the same time all of other human associated Streptomycesare separated from each other, so it is not very specific for the TR1341, that should be clearly stated in the manuscript. For the other pathogenic Actinobacteria, based on dataset it is also not uncommon that the strain is separated from them, as they belong to different genera and just few strains of these genera were included which is the cause they were grouped. Perhaps changing the data presentation on Figure 1 so it shows evolutionary distance would improve the quality of the figure and message it delivers.

Answer: We added the branch length values in the Figure 1 and added a comment regarding the position of the human-associated streptomycetes at the end of the paragraph (l. 384-390). OrthoFinder uses not only single copy orthologous genes, but also the full set of orthologous genes for phylogenetic inference. Due to the algorithm used for tree inference (STAG), the branch lengths correspond to the average number of substitutions per site across a large range of gene families [70]. Therefore, the resulting tree can be used directly in downstream analysis such as ancestral state reconstruction and calibration.

  • Line 366 – ‘…made by [27].’ I suggest changing it to ‘…made by Herbrik, et al.[27].’

Answer: We changed the text according to the reviewer’s suggestion (l. 381).

  • Line 397 – it is not clear to me what the information in brackets is refereeing too, specifically ‘…(GYM media, samples 1 and 2)’ – please specify that in the text.

Answer: We added the reference to Figure 3b, where you can find the values of sample/replicate 1 and 2.

  • Line 398 – what the author meant by ‘…has only one biosynthetic arm…’?

Answer: The biosynthetic cluster for actinomycin was first characterized in Streptomyces chrysomallus, where the main genes are flanked by two almost mirrored regions containing the rest of the biosynthetic genes (Ref nr 89-90). In that paper, the authors decided to call these regions “arms” (we guess they resemble the arm of a person with the main genes being the body). Here, we decided to use the same terminology as suggested by these authors, so that it might be easier for the reader to navigate the different studies.

  • Line 415 – what the number in brackets refers to, is it the contig mentioned in the sentence or reference sequence of NRPS involved in the synthesis of rhizomides in another strains, specify please.

Answer: It’s the accession number of the MiBIG database. We added the specification in the text.

  • Lines 594 – 596 – authors state that low sequence similarity points to different function which in my opinion is a slight overstatement. It might be true, however does not have to be as the function of proteins is determined by theirs structure, and even though the sequence defines structure, there is still a possibility that proteins with low sequence similarity will have highly similar structure. That is usually the case of the proteins under strong evolutionary pressure, e.g. in vertebrates immune system. If the protein is involved in the pathogenicity of the strain, it would be reasonable to highly diversify its structure, so the chances to be recognized by the host immune system are minimized. I suggest rephrasing the sentence so it discusses on the options and is less suggestive.

Answer: We have changed the segment to be read less like an affirmation and more like a possibility. We agree with the reviewer that sequence similarity is not the only defining trait that should be look at when talking about protein function and that the 3D structure of the protein is the one that defines the functioning of the protein. In this case we think is better not to affirm that the protein is an internalin but to let the possibility open for it to be something else because we don’t have a 3D model. This uncertainty in the annotation is denoted by calling the protein a putative internalin and not just an internalin (l. 649-653).  

Reviewer 2 Report

The manuscript of Lara et al. is a continuation of the paper of Herbrík et al. 2020

(Front. Microbiol, 10, 3028, doi:10.3389/fmicb.2019.03028) and is focused in the comparative phylogenetic analysis of the Streptomyces sp.TR1341 strain with environmental and human pathogen Streptomyces strains.

The manuscript is well written and presents an extensive comparison of different types of genes. 

I only have minor suggestions and comments:

Figure 1: please include the indication of the corresponding isolation source for the strain Streptomyces sp.TR1341 (the color circle that the authors have  indicated in the other strains)

Table2 and table S3: the authors described (Herbrík et al. 2020) that this strain produces filipin. However, the cluster of this molecule is not indicated in the mentioned tables. Please clarify.

The figure S4 is not referred in the manuscript. Please make a reference to it.

Figure S6, please include the names of the compared microorganisms in the circle (as in figure S4).

Figure 7. It is hard so see the numbers in the black part of the figure (antibiotic and iron metabolism parts). Please use a bigger size.

The names of the microorganisms are not in italics in the references part; please correct.

Author Response

The manuscript of Lara et al. is a continuation of the paper of Herbrík et al. 2020

(Front. Microbiol, 10, 3028, doi:10.3389/fmicb.2019.03028) and is focused in the comparative phylogenetic analysis of the Streptomyces sp.TR1341 strain with environmental and human pathogen Streptomyces strains.

The manuscript is well written and presents an extensive comparison of different types of genes. 

I only have minor suggestions and comments:

Figure 1: please include the indication of the corresponding isolation source for the strain Streptomyces sp.TR1341 (the color circle that the authors have  indicated in the other strains)

Answer: We added the corresponding red dot near the position of strain TR1341.

Table2 and table S3: the authors described (Herbrík et al. 2020) that this strain produces filipin. However, the cluster of this molecule is not indicated in the mentioned tables. Please clarify.

Answer: As suggested, we added filipin to the table. Originally, we reported in those tables the exact output of the antiSMASH prediction as displayed in their summary table. As most similar BGC for Region1.3, antiSMASH put as first hit Pentamycin and as second hit with the same percentage of genes showing similarity, there is filipin. However, the second hit is not shown in antiSMASH overview table. Pentamycin is a synonym of fungichromin, which is another polyene, very similar to filipin. Both compounds are produced by TR1341 as shown in Herbrik et al., 2020.

The figure S4 is not referred in the manuscript. Please make a reference to it.

Answer: The first reference to figure S4 is at l. 380.

Figure S6, please include the names of the compared microorganisms in the circle (as in figure S4).

Answer: As suggested, we added the labels.

Figure 7. It is hard so see the numbers in the black part of the figure (antibiotic and iron metabolism parts). Please use a bigger size.

Answer: We have improved the readability with larger font in the Figure 7.

The names of the microorganisms are not in italics in the references part; please correct.

Answer: It is corrected. The names of the microorganisms are now in italics in the reference list.

Round 2

Reviewer 1 Report

Dear Authors,

All the issues has been addressed and missing explanations provided. I have no more comments.

Kind regards,
Reviewer